# Investigations of Optical Functions and Optical Transitions of 2D Semiconductors by Spectroscopic Ellipsometry and DFT

**DOI:** 10.3390/nano13010196

**Published:** 2023-01-01

**Authors:** Honggang Gu, Zhengfeng Guo, Liusheng Huang, Mingsheng Fang, Shiyuan Liu

**Affiliations:** 1State Key Laboratory of Digital Manufacturing Equipment and Technology, Huazhong University of Science & Technology, Wuhan 430074, China; 2Optics Valley Laboratory, Wuhan 430074, China; 3Innovation Institute, Huazhong University of Science and Technology, Wuhan 430074, China; 4School of Optical and Electronic Information, Huazhong University of Science and Technology, Wuhan 430074, China

**Keywords:** optical transitions, optical functions, monolayer WS_2_, spectroscopic ellipsometry, first-principle calculations

## Abstract

Optical functions and transitions are essential for a material to reveal the light–matter interactions and promote its applications. Here, we propose a quantitative strategy to systematically identify the critical point (CP) optical transitions of 2D semiconductors by combining the spectroscopic ellipsometry (SE) and DFT calculations. Optical functions and CPs are determined by SE, and connected to DFT band structure and projected density of states via equal-energy and equal-momentum lines. The combination of SE and DFT provides a powerful tool to investigate the CP optical transitions, including the transition energies and positions in Brillouin zone (BZ), and the involved energy bands and carries. As an example, the single-crystal monolayer WS_2_ is investigated by the proposed method. Results indicate that six excitonic-type CPs can be quantitatively distinguished in optical function of the monolayer WS_2_ over the spectral range of 245–1000 nm. These CPs are identified as direct optical transitions from three highest valence bands to three lowest conduction bands at high symmetry points in BZ contributed by electrons in S-3*p* and W-5*d* orbitals. Results and discussion on the monolayer WS_2_ demonstrate the effectiveness and advantages of the proposed method, which is general and can be easily extended to other materials.

## 1. Introduction

Ultra-thin two-dimensional (2D) semiconductors, representatively including the 2D transition metal dichalcogenides (TMDCs) [1], phosphorene [2], 2D hybrid perovskites [3], etc., have attracted extensive scientific attentions during the past two decades. Two-dimensional semiconductors have atomic thicknesses in the vertical direction, and the reduced dimensionality introduces various unique optical and electrical properties different from those in their 3D counterparts, such as the tunable bandgap [4,5,6,7], giant optical anisotropy [8,9], spintronics [10], valleytronics [11], high mobility and on/off ratio [2,12], etc. On the basis of these outstanding characteristics, 2D semiconductors have been widely applied to various optoelectronic devices, such as field-effect transistors [13,14], light-emitting diodes [15], solar cells [16], photodetectors [17], sensors [18], etc. With continuous physics explorations and technique breakthroughs in large-scale single crystal preparation [19], the 2D semiconductor has been one of the most promising candidates for the next-generation photonics and optoelectronics, especially for the angstrom-node integrated circuits [20], flexible electronics [21], etc. Systematical characterization and revelation of the light–matter interactions in 2D semiconductors provide the basis to clearly understand and regulate the optical and electrical properties, and to achieve practical applications to devices [22,23].

The band structures of atomic 2D materials are quite different from those of the bulk materials, which is the physical origin of the specific optical responses [24]. Fist-principle calculations are important means to predict the fundamental band structures and optical properties of 2D materials, and a lot of fist-principle studies have focused on the band structures and optical responses of 2D TMDCs [25,26,27,28,29]. The critical point (CP) optical transitions are essential physical characteristics for semiconductors, which are closely related to the band structures of materials and can directly reflect the light–matter interactions, such as the light absorption and light emission [22,23,24,30,31,32]. Therefore, CP optical transitions build a bridge between the optical responses and the band structures of 2D semiconductors. Accurately determining the detailed information of the optical transitions, such as the occurring positions, and the involved energy bands and carrier types can not only help us to physically understand novel photoelectric phenomena and properties, but also provide a quantitative guide for the property regulation and optimal design of materials and related devices [22,33,34,35,36]. For example, the CP optical transitions can help us to evaluate the absorption/emission feature peaks in novel 2D materials, and further choose proper strategies to enhance the light absorption/emission to achieve high-performance optoelectronic devices, such as the light-emitting diodes and solar cells [22].

Optical functions, including the complex dielectric function, refractive index, and optical conductivity, are regarded as the optical fingerprints of materials, and they play important roles in the investigation of optical and dielectric properties of materials and the architecture design of devices [31]. For example, the dielectric functions describe the impact of the electromagnetic field on the matter, and the real and imaginary dielectric functions represent, respectively, the modulation of electromagnetic field phase indicating the dielectric polarization capability and the modulation of the electromagnetic field amplitude indicating the dielectric loss capability. They play important roles in the design of electronic devices such as transistors, diodes, capacitors, etc. It is a common practice to identify the optical transitions by combining the experimentally determined optical functions with the theoretically calculated band structures [4,31,32]. Although some well-known techniques, such as the angle-resolved photoemission spectroscopy (ARPES) [37], scanning tunneling microscopy and spectroscopy (STM/STS) [38], and photoluminescence (PL) [30], can provide some information about optical transitions by probing the band structures, they fail to obtain the optical functions. Optical functions of 2D semiconductors can be experimentally determined by optical spectroscopic techniques, including the conventional absorption/reflectance/transmission (ART) spectroscopy and its derivative techniques [39,40,41,42], and the spectroscopic ellipsometry (SE) [4,5,6,7,8,9,31,43,44,45]. Compared with the ART-based methods that need additional physical constraint functions, e.g., the Krames–Kronig relation and Fresnel equations, to extract the complex optical functions [39,42], the SE has been proved to be a powerful, self-consistent and highly sensitive tool to characterize the optical and dielectric properties of nanomaterials [31]. The SE provides approximately twice the amount of information of that of ART spectroscopy by detecting optical responses of two orthogonal polarization directions, and can directly achieve the complex optical functions without any additional physical constraint functions. Although the SE faces big challenges in dealing with 2D materials due to the extremely limited optical path [46], many researchers have devoted their efforts to address this issue and developed specific SE for 2D materials [47,48]. However, in reviewing the published studies, most of them focused on determining the basic optical functions and investigating their layer-dependent or field-regulated optical and dielectric properties of 2D materials by SE [4,5,6,7,8,9,31,33,34,35,36,37,38,39,40,41,42,43,44,45], there lacks a systematic method to comprehensively investigate the detailed information about the CP optical transitions of 2D semiconductors.

In this paper, we aim to provide a systematic and quantitative strategy to investigate the optical functions and optical transitions of 2D semiconductors by connecting the experiments with the density functional theory (DFT). Based on this objective, we propose a general method to systematically identify the CP optical transitions of 2D semiconductors by combining the SE and first-principle calculations. The physical basis is that feature peaks in the SE-determined optical functions correspond to the CP optical transitions of materials, and the differential band structure, namely the difference between the conduction bands and valence bands, covers all possible energy levels for the optical transitions of materials. In addition, extreme points of the differential band structure correspond to the Van Hove singularities, which can help us to determine the exact positions of the CP optical transitions in the band structure. Firstly, optical functions of the 2D semiconductor are determined by the SE. Then, the CPs in the optical function spectra can be determined by using the CP analysis method. Finally, the CP optical transitions can be identified by connecting the CP analysis results of the experimentally determined optical functions and the first-principle calculated band structure and projected density of states (PDOS) using the equal-energy lines (EELs) and equal-momentum lines (EMLs). With the proposed method, we can determine the energy and spatial positions in Brillouin zone (BZ), and the involved energy bands and carrier types of the optical transitions occurring at the CPs of 2D semiconductors. In the results and discussion section of the paper, the monolayer WS_2_ is taken as an example to demonstrate the effectiveness and verification of the proposed method. Results and findings about the optical functions and CP optical transitions of the monolayer WS_2_ are vital physical knowledge for the understanding and regulation of its optoelectronic properties and application promotion in related devices. In addition, the proposed method can be easily extended to other materials.

## 2. Theory and Methods

### 2.1. General Strategy to Investigate Optical Functions and Optical Transitions of 2D Semiconductors

The spectroscopic ellipsometry (SE) detects the polarized light–matter interactions via polarization modulation and demodulation devices, and can provide more abundant information about the sample by introducing an additional dimension of the light, namely the polarization. The SE has been developed as a powerful and standard tool for the characterization of basic optical functions and film thicknesses of nanomaterials and nanofilms. The typical SE analysis contains two basic issues, namely the forward model and the inverse extraction. The sample information can be forward modeled within the complex reflection coefficient and then inversely extracted from the measured ellipsometric angles, as described by [49]
(1)$$\rho \left(\varepsilon ,d\right)=\frac{r_{p}\left(\varepsilon ,d\right)}{r_{s}\left(\varepsilon ,d\right)}=\tan\Psi \exp(i\Delta ),$$
where *ρ* is the complex reflection coefficient, *r*_p_ and *r*_s_ refer to the reflection coefficients of the p-polarization and s-polarization light components, respectively, Ψ and Δ are the amplitude ratio angle and the phase difference angle, respectively, *ε* = *ε**_r_* − *i**ε**_i_* is the dielectric function of the material (here, *ε*_*r*_ and *ε**_i_* are the real part and imaginary part of the dielectric function, respectively), and *d* represents the structure parameters (usually the film thickness) of the sample.

In the forward modeling procedure, usually two models should be constructed, including an optical stacking model to describe the sample and a parametric oscillator model to physically describe the dispersive dielectric functions of the involved materials [44,48,49]. Based on these models, computational electromagnetics methods, such as the transfer matrix method, can be applied to theoretically calculate the complex reflection coefficient. Then, the experimentally measured ellipsometric angles are inversely fitted by the theoretically calculated ones, and the sample information can be finally extracted [4,49]. For ultrathin 2D materials, the dielectric functions can be also analytically determined from the measured ellipsometric angles with the thickness estimated by the layer number and the spacing between two adjacent layers [31,48]. The complex refractive index can be directly converted from the dielectric functions according to the relation *ε* = *N*^2^ = (*n* − *i**k*)^2^, where *n* and *k* are the refractive index and the extinction coefficient, respectively. Further, the complex optical conductivity of a 2D material can be calculated either from the complex refractive index (or dielectric functions) by the classic slab model or directly from the measured ellipsometric angles by the surface current model [50]. Therefore, we can determine all the three basic complex optical functions of a 2D semiconductor via the SE, which consequently enables us to quantitatively investigate the optical and dielectric properties.

In general, there are some feature peaks in the optical functions over the optical band, which correspondingly reflect different optical transitions in the band structure of the material. Accurately identifying these feature peaks can help us to quantitatively investigate the optical transitions and further reveal the light–matter interactions in 2D materials. We can identify the feature peaks in optical function spectra by using the well-known critical point (CP) analysis method. In the CP analysis, the second derivative of the dielectric functions with respect to the photon energy is calculated and then fitted by the following formulas [32]:(2)$$\frac{d^{2}\varepsilon }{dE^{2}}=\{\begin{array}{l} m(m-1)Ae^{i\phi }{\left(E-E_{0}+i\Gamma \right)}^{m-2};\;m\neq 0\\ Ae^{i\phi }{\left(E-E_{0}+i\Gamma \right)}^{-2};\;m=0 \end{array},$$
where *A* is the amplitude, ϕ is phase angle, *E*_0_ is the center energy, Γ is the broadening of the peak, and *m* is the dimensionality of the CP. The CP dimensionality depends on the band structure, and *m* = 1/2, 0, −1/2, and −1 correspond to the three-dimensional, two-dimensional, one-dimensional, and excitonic CPs, respectively. Therefore, by taking advantage of the CP analysis method, the central energies and the dimensionalities of the CPs can be precisely determined.

On the other hand, the optical and dielectric properties of 2D materials can be theoretically predicted by the DFT. Especially the band structure, partial density of states (PDOS) and optical functions can be calculated with the help of various first-principle tools, such as Vienna Ab initio Simulation Package (VASP), Cambridge sequential total energy package (CASTEP), etc. Based on the first-principle calculations, the differential band structure (*E*_C−V_) can be obtained by calculating the energy difference between the conduction bands (*E*_C_) and valence bands (*E*_V_), i.e., *E*_C−V_ = *E*_C_ − *E*_V_. Further, the joint density of state (JDOS) can be defined with the differential band structure (*E*_C−V_) as provided by
(3)$$JDOS(E_{C-V})=\frac{1}{4\pi^{3}}\int \frac{dS_{K}}{\left|\nabla_{K}E_{C}{}_{-V}\right|},$$
where *S_K_* is the constant energy surface defined by *E*_C−V_ = constant. According to the microscopic theory of dielectric functions [51], the JDOS determines the optical transition probability. It can be seen from Equation (3) that the JDOS possesses singularities at points when ∇*_K_E*_C−V_ = 0. These points are known as the Van Hove singularities, just corresponding to the tangent points in the differential band structure (*E*_C−V_), which can be regarded as the transition positions.

Finally, we can systematically investigate the CP optical transitions by combining the SE, CP analysis, and first-principle calculations. As schematically shown in Figure 1, the band structure (Figure 1a), the PDOS (Figure 1b), the calculated differential band structure (Figure 1c), and the CP analysis results (Figure 1d) are drawn in sequence. Specific information about the CP optical transitions, including the occurring position in the energy and momentum space, and the involved energy bands and carrier types can be identified by connecting these results via equal-energy lines (EELs) and equal-momentum lines (EMLs). It should be noted that since the SE cannot obtain the fundamental bandgap, the bandgap used in this work is referred to the optical bandgap, which is usually much smaller than the fundamental bandgap (i.e., the quasiparticle bandgap) considering the excitonic binding energy. As summarized in Figure 1, the proposed method to determine the CP optical transitions of 2D semiconductors can be operated in the following steps.

(1) Firstly, at the center energies of the CPs, EELs are drawn across the second derivative of dielectric functions (i.e., CP analysis results) and the differential band structure. The tangent points between the EELs and the differential energy bands are identified, which can be regarded as the occurring positions of the CP optical transitions. In this procedure, the energy and momentum positions in the Brillouin zone (BZ) can be determined for the CP optical transitions.

(2) Then, across the tangent points, EMLs are drawn through the differential band structure and band structure diagrams. The specific energy bands, including valence bands (VB) and conduction bands (CB) involved in the CP optical transitions, can be determined. In the case of light absorption, the electrons are transformed from the ground state to the excited state by absorbing photons, where the transition is from VB to CB, known as the excitation transition, while in the case of light emission, the electrons fall back from the excited state to the ground state by releasing photons, and the transition is from CB to VB, known as the relaxation transition.

(3) Finally, across the intersection points between the EMLs and the involved energy bands, EELs are drawn again through the band structure and the PDOS. Specific carrier types involved in the CP optical transitions can be determined by distinguishing the peak points around the intersection points between the EELs and the PDOS curves of electrons on specific orbits.

In summary, by using the proposed method, we can systematically investigate the CP optical transitions of 2D semiconductors. Detailed information, including the energy and momentum positions in the BZ, and the involved energy bands and carrier types of the CP optical transitions can be revealed, which provides basic physical knowledge of a novel material to help not only understand its optical and dielectric phenomena, but also guide the property control and design of related devices. It should be noted that there are various functions that can be adopted to perform the DFT calculations. Although different DFT functions may result in different bandgaps, the dispersion relations of the band structures are similar to each other. Before analyzing the CP transitions using the proposed approach, a direct-current offset (i.e., a constant) should be added to the differential band structure to make the DFT calculated bandgap match the SE measured one. Then, the CP optical transitions above the bandgap can be specifically identified.

In the rest of this paper, the monolayer tungsten disulfide (WS_2_) is taken as a representative example to demonstrate the effectiveness and advantages of the proposed method in identifying the CP optical transitions of 2D semiconductors. It is noteworthy that the proposed method itself is general and can be applied to investigate the optical transitions of various kinds of materials.

### 2.2. Monolayer WS_2_ Examinations: A Typical Example

In the following discussion, the monolayer WS_2_ is taken as a typical example to verify the above methods. In this section, detailed examinations about the monolayer WS_2_, including the sample preparation and characterization, SE measurements, and first-principle calculations, are presented.

The large area single-crystal triangular monolayer WS_2_ on the sapphire (Al_2_O_3_) substrate, whose maximum side length was 65 µm, was provided by SixCarbon Technology Shenzhen. The triangular monolayer WS_2_ sample was prepared by the Chemical Vapor Deposition (CVD) process. In order to prove the high quality of our sample, a series of characterizations was performed on the monolayer WS_2_ sample. The surface morphology of the CVD monolayer WS_2_ sample was examined by the atomic force microscopy (AFM) (Bruker Dimension Icon) and scanning electron microscope (SEM) (Zeiss GeminiSEM 300). In addition, in order to examine the crystal quality of the WS_2_ sample, Renishaw In-Via Reflex spectrometer was applied to obtain the Raman and photoluminescence spectra with an excitation wavelength of 532 nm at the room temperature of approximately 20 °C.

The triangular monolayer WS_2_ sample was investigated by a customized micro-spot spectroscopic ellipsometer (Wuhan Eoptics Technology Co., Wuhan, China). The ellipsometer is based on the dual rotating compensator principle [52], consisting of a light source, a polarization state generator (PSG), a sample stage, a polarization state analyzer (PSA), a detector, and an auxiliary CCD image system. The light source is a deuterium and quartz–tungsten–halogen combined light source. The PSG contains a linear polarizer followed by a multi-waveplate as the first rotating compensator, while the PSA contains a linear polarizer known as the analyzer following another multi-waveplate as the second rotating compensator [53]. The detector is a customized spectrometer. With these key components, the applicable wavelength range of the ellipsometer covers 245–1000 nm (i.e., the energy range of 1.24–5.06 eV). The SE measurements were performed at an incident angle of 60°. The micro-spot size of the probe light can be as small as approximately 50 µm to detect tiny samples, and a CCD camera is integrated on the ellipsometer to help us determine the test points (i.e., the micro-spot positions of the probe light). In addition, to further confirm the reliability of SE measurements and analysis, we have performed additional transmittance and absorbance measurements on the monolayer WS_2_ sample over the spectral range of 380–830 nm by using a commercial spectrophotometer (SolidSpec-3700, Shimadzu Co., Ltd., (Kyoto, Japan)).

The band structure, partial density of states (PDOS) and dielectric functions of the monolayer WS_2_ were calculated with the help of Vienna Ab initio Simulation Package (VASP v5.4.1) [54]. In the calculations, initial values of WS_2_ lattice constants are adopted from experimental lattice parameters, i.e., the space group is P63/mmc, the lattice constants of a = b = 3.18 Å, and c = 12.60 Å [55]. A vacuum slab of 18 Å is applied between the structural layers to remove the interlayer interaction in the monolayer WS_2_. At the density function theory (DFT) level, the Perdew–Burke–Ernzerhof (PBE) pseudopotential based on projector augmented wave (PAW) was used in the whole calculation process [56,57]. Based on the convergence tests, the plane wave cutoff energy was set to 600 eV, and the Brillouin zone (BZ) used an 8 × 8 × 1 Monkhorst-Pack K-point grid. In the structure optimization, the convergence criterions of the force and energy were set to 0.01 eV/Å and 10^−5^ eV, respectively. For the PDOS calculations of the monolayer WS_2_, the BZ was sampled by a denser 12 × 12 × 1 K-point grid to obtain the ground state charge density. Except for PBE pseudopotential, the HSE06 pseudopotential [58] was also applied to obtain the reasonable optical bandgap, which is usually comparable with the SE experimental value and therefore helps to perform the following analysis of the CP optical transitions.

The optical properties of the monolayer WS_2_ were calculated independently by partial approximation (IPA) and G_0_W_0_-Bethe−Salpeter equation (G_0_W_0_-BSE) approaches [59,60,61]. For the IPA calculations, the plane wave cutoff energy was set to 600 eV and BZ was sampled by a 12 × 12 × 1 K-point grid. In order to involve the excitonic effects, the G_0_W_0_-BSE approach was applied. In G_0_W_0_-BSE calculations, a stricter energy convergence criterion of 10^−8^ eV of the ground state was used. Firstly, the converged PBE wavefunction and eigenvalues obtained in ground calculation were used in a single GW iteration (G_0_W_0_) calculation [62], during which the quasiparticle energy was recalculated during the G_0_W_0_ process and the wave function remained at the PBE level. Then, in order to consider the excitonic effects, the quasiparticle energy and wave function were brought into the Bethe−Salpeter equation (BSE) calculation [63], and the Tamm–Dancoff approximation was used to obtain the G_0_W_0_-BSE spectrum. Among them, the eight highest valence bands and the eight lowest conduction bands were used as the basis for the calculation of excitonic intrinsic states. The spin–orbit coupling (SOC) effect was considered throughout the calculations.

## 3. Results and Discussion

As a representative transition metal dichalcogenide (TMDC), WS_2_ has trigonal prismatic structure and belongs to the hexagonal space group P63/mmc [55]. The inset in Figure 2a shows a schematic diagram of the lattice structure of the monolayer WS_2_, demonstrating that one tungsten (W) layer is sandwiched by two sulfur (S) layers. The PL spectrum in Figure 2a indicates that the monolayer WS_2_ has a significant emission peak at 2.003 eV, which is consistent with the previous study [64]. The monolayer WS_2_ is a direct bandgap semiconductor, and the direct optical transition that occurs at the first exciton peak just corresponds to the direct optical bandgap of the monolayer WS_2_. Therefore, the PL results indicate that the direct optical bandgap of the monolayer WS_2_ is 2.003 eV. In addition, it can be seen that the PL band shows inhomogeneous broadening, which may be induced by the sapphire substrate (by deformation in particular) of the monolayer WS_2_ [65]. Figure 2b shows the Raman spectrum of the monolayer WS_2_, and the spectrum exhibits two phonon modes, namely E^1^_2g_ mode at 354.1 cm^−1^ and A_1g_ mode at 417.5 cm^−1^, suggesting the high purity of the monolayer WS_2_ sample [66]. Figure 2c shows the absorption and transmission spectra, which clearly demonstrates that there are four feature peaks labeled with A–D. These feature peaks correspond to specific optical transitions in the monolayer WS_2_, and their energy positions are summarized in Table 1. Especially, the feature peak A just corresponds to the optical transition above the optical bandgap, and its center energy is about 2.031 eV, which highly agrees with the PL bandgap 2.003 eV. In addition, Figure 2d displays an AFM image of the triangular monolayer WS_2_ crystal. The AFM picture shows that the thickness of monolayer WS_2_ is around 0.81 nm, and the triangular structure is well presented. The SEM images shown in Figure 2e,f also obviously indicates that the morphology structures of the monolayer WS_2_ samples show clear triangular shapes, indicating the grain size of the single-crystal monolayer WS_2_ is from 40 µm to 65 µm. Figure 2f illustrates an example grain size of 65 µm for the single-crystal monolayer WS_2_. These characterization results indicate the high quality of our triangular monolayer WS_2_ sample, which is the reliable foundation for the next SE measurements.

Figure 3a schematically illustrates the fundamental structure of the spectroscopic ellipsometer adopted in this study, which consists of a light source, a polarization state generator (PSG), a sample stage, a polarization state analyzer (PSA), a detector, and an auxiliary CCD image system. The SE is a fast, accurate, non-contact and self-consistent optical technique, and it is widely applied to study the inherent optical properties of nanomaterials and nanostructures. The SE measurement is performed at an incidence of *θ* = 60°, and the inset of Figure 3a is an image obtained by the CCD, which shows the light spot irradiated on the sample. An optical stacking model, consisting of three parts, namely the air, the WS_2_ layer and the sapphire substrate, is constructed to describe the sample. In addition, the dielectric functions of the monolayer WS_2_ are parametrically described by combining two Gaussian, two Cody–Lorentz and four Lorentz oscillators over the whole concerned spectral range [49]. Figure 3b shows the measured and best-fitting ellipsometric spectra (with Psi for Ψ and Delta for Δ) of the monolayer WS_2_ sample. Based on the constructed optical and dielectric models, the theoretical ellipsometric spectra (solid lines) are calculated by the transfer matrix method, and they are in good agreement with the measured ellipsometric angles (solid circles) as shown in Figure 3b. The best-fitting thickness of the WS_2_ layer is 0.64 nm, which is highly consistent with the theoretical value of 0.62 nm [55], and also close to the AFM result presented in Figure 2d.

With the SE measurement and analysis, three pairs of basic optical functions, i.e., the complex dielectric functions (*ε* = *ε**_r_* − *i**ε*_*i*_), complex refractive index (*N* = *n* − *i**k*), and complex optical conductivity (*σ* = *σ*_*r*_ − *i**σ*_*i*_), can be simultaneously determined for the monolayer WS_2_. Although these three complex parameters can be converted into each other, they refer, respectively, to different physical meanings and describe different properties of the material from different aspects. The dielectric functions describe the dielectric polarization and dielectric loss properties of the medium. The real part (*ε*_*r*_) of dielectric function represents the modulation of the electric field phase, which indicates the storage ability of the medium to the electric field energy, and describes the conversion process between the potential energy of the electric field and Coulomb potential energy of the medium, while the imaginary part (*ε*_*i*_) of the dielectric function represents the modulation of the electric field amplitude and describes the loss energy of the electric field, and most of the loss energy is converted into free electron kinetic energy and thermal energy. The complex refractive index indicates the ability of a medium to refract and dissipate the light. The refractive index (*n*) is defined as the ratio of the speed of light in vacuum to the phase velocity in the medium, which represents the refraction ability of light in the medium, while the extinction coefficient (*k*) represents the attenuation of the incident light when it propagates through the medium, including the absorption loss and other inelastic scattering loss, etc. The optical conductivity provides the relationship between the induced current density in a material and the magnitude of the inducing electromagnetic field, which is an essential property of a material for the development of relevant optoelectronic devices, especially photoconductors, photovoltaics, photodetectors, etc. Specifically, in this paper, the optical conductivity is referred to the photoconductivity, and its real part and imaginary part are directly related to the photocurrent and photoresistance of the material, respectively. The monolayer 2D material can be treated as an infinitely thin sheet with light-induced surface currents [50,67,68]. In this case, the real part (*σ*_r_) of complex optical conductivity describes the ability to generate conducting photocurrents, and it reflects the energy loss component caused by the conduction current excited by the light, while the imaginary part (*σ*_i_) of complex optical conductivity describes the ability to generate displacement photocurrents excited by the incident light, and it reflects the energy storage capacity of the material to the incident light.

Figure 4 demonstrates the dielectric function, complex refractive index and complex optical conductivity spectra of the monolayer WS_2_ over the energy range of 1.24–5.06 eV (245–1000 nm) determined by the SE. It can be obviously observed from Figure 4 that the optical function spectra of the monolayer WS_2_ exhibit six significant feature peaks as labeled by uppercases A–F in the extinction coefficient spectrum. It should be noted that the peaks in the extinction coefficient spectrum are completely consistent with those in the imaginary dielectric function spectrum and those in the real optical conductivity spectrum, as indicated by the sky-blue bands in Figure 4b,d,f. These feature peaks indicate the peak positions of the light absorption or light emission induced by the CP optical transitions in the monolayer WS_2_. For three low-energy feature peaks A–C, they have relatively smaller amplitudes and narrower broadenings compared with high-energy feature peaks D–F. Consequently, we can roughly speculate that the optical transitions corresponding to the feature peaks A–C should be simple and unambiguous, since complex optical transition procedures usually require high-energy photons to excite, while the high-energy feature peaks D–F are probably resulted from the combination of several different optical transitions, such as the electron and excitonic transitions, the defect transitions, etc.

In order to accurately determine and analyze the information of these feature peaks (A–F) of the monolayer WS_2_, the CP analysis is further performed on the dielectric function spectra. In the CP analysis, the second derivative of the SE measured dielectric function with respect to photon energy is fitted by a theoretical parameterized formula as provided in Equation (2). The parameters corresponding to the best-fitting results of the CP analysis are summarized in Table 2. It can be seen that the theoretical curves fit very well with the second derivative of the experimental dielectric function. By taking advantage of the CP analysis method, the central energies, shapes and dimensionalities of the CPs can be precisely determined. For the monolayer WS_2_, the dimensionalities of the best-fitting CP curves are *m* = −1, indicating all these six CPs A–F correspond to excitonic-type optical transitions.

To further confirm the reliability of the SE results and CP analysis, we have performed additional transmittance and absorbance measurements on the monolayer WS_2_ sample using a commercial spectrophotometer, as shown in Figure 2c. It can be seen that there are four feature peaks (labeled with A–D) on the absorbance and transmittance (A and T) spectra, which correspond to the CPs A–D on the SE determined optical function spectra. Table 1 summarize the comparison of CP center energies *E*_0_ obtained from the SE, A and T, PL, as well as the first-principle calculations. It can be observed that these results highly agree with each other, indicating the reliability of our SE measurements and analysis. It should be noted that the PL can only provide a strong peak around the direct optical bandgap of the monolayer WS_2_. In addition, the optical bandgap measured by the PL is generally a little smaller than those determined by the SE or A and T, which can be confirmed by center energies of peak A shown in Table 1.

The difference in the position of the peak A for PL, absorption and SE results may be induced by three reasons. The first reason is the different working mechanisms of these techniques. The PL detects the light emission under a high-energy radiation laser, which corresponds to the relaxation transition from the excited to the ground state in the material, while the absorption and SE detects the light absorption, where the electrons are excited from the ground state to the excited state by absorbing photons, and corresponds to the excitation transition in the material. In general, the amount of energy of relaxation transition is a little smaller than that of the corresponding excitation transition, which makes center energy position of peak A in PL spectrum smaller than that in absorption and SE spectra. The second reason is that the energy position of peak A in the SE dielectric function spectra is determined by a rigorous CP analysis, namely by fitting the second derivative of the dielectric function, while the energy positions of peak A in the PL and absorption spectra are roughly determined by the maximum values. These two different data processing methods may introduce some difference in the position of the A peak. The last possible reason is the measurement errors and conditions in the three different instruments. The difference between the A and T peak positions and SE peak positions for high-energy peaks B–D shown in Table 1 is similar to that of peak A. The above explanation about the possible reasons can be extended to these high-energy peaks B–D.

It should be noted that the imaginary dielectric function (*ε*_2_) and extinction coefficient (*k*) remain non-zero below the bandgap as shown in Figure 4, which are non-physical artifacts. We think that two possible reasons may induce the extrinsic absorption in the infrared region. One is that the SE spot size is approximately the same, even larger than the grain size, which can be confirmed by Figure 3a. In this case, the SE detected signal may involve scattering intensity especially from the grain edge, and these scattering signals mainly result in infrared absorption over the low energy range. Another reason is that the CVD monolayer WS_2_ sample may contain some defects and impurities, such as the transfer residues, and these imperfect features may also introduce extrinsic absorption especially in the low energy range (i.e., the infrared region). In addition, the poor ellipsometry response in the infrared region leads to low signal-to-noise ratio of the measured ellipsometric data as shown in Figure 3b, which maybe further exacerbates the analysis of the optical functions. Nonetheless, the main results concerned and discussed in this work, such as the energy levels of the CPs, have been confirmed by other techniques, including the first-principle results and A and T spectra, as shown in Table 1.

Further, we performed first-principle calculations on the band structure, PDOS and optical functions of the monolayer WS_2_ to compare them with the experimental results. The calculations were performed under different approximations, including the HSE06, PBE, IPA and G_0_W_0_-BSE, to provide a thorough understanding of the CP optical transitions in WS_2_ from the theoretical perspective. Figure 5a shows the HSE06 band structure of the monolayer WS_2_. The direct optical bandgap *E*_g_ at K point is 2.03 eV, which highly agrees with the PL, SE and A and T results in this paper and previous experiment value [69]. When the SOC effect is taken into consideration, the energy splitting Δ*E* that occurs in the VB maximum at K point is due to the absence of inversion symmetry [70], and the splitting energy Δ*E* is approximately 379 meV. The PDOS of the monolayer WS_2_ under the HSE06 level is illustrated in Figure 6b, indicating that the major contribution to the PDOS originates from the electrons in 3*p* orbitals of S atoms, 5*d* and 6*s* orbitals of W atoms. In addition, the PBE approximations have also been performed and the PBE results are presented in Figure 5d. For the PBE band structure, the band dispersions (band energy vs. momentum space) in band structure are extremely similar to the HSE06 results, but the PBE functional underestimates the bandgap, which is approximately 1.54 eV, much smaller than the HSE06 bandgap 2.03 eV.

Figure 5b,c and Figure 5e,f display the dielectric function spectra of the monolayer WS_2_ calculated under G_0_W_0_-BSE and IPA approximations, respectively. The excitonic effects have been taken into consideration in the G_0_W_0_-BSE calculations, while the IPA does not involve the excitonic effects, and that is why peaks A and B in Figure 5e,f are weaker than the G_0_W_0_-BSE results in Figure 5b,c. In addition, the IPA results are based on the PBE functional, so the energy positions of A and B are smaller than the results of G_0_W_0_-BSE. It can be seen that the shapes of the theoretically calculated dielectric function spectra are similar to the experimentally determined dielectric function spectra as provided in Figure 4. Additionally, around the center energies of the experimentally determined CPs as summarized in Table 1, the G_0_W_0_-BSE theoretically calculated dielectric function spectra also exhibit six corresponding feature peaks (A–F) as presented in Figure 5c. For example, the center energies of peak A and B in Figure 5c are approximately 2.16 eV and 2.50 eV, which are close to the energy positions of CP A and B (2.02 eV and 2.41 eV) in Figure 4d. As a consequence, the agreement between the experimental and theoretical results demonstrates the reliability of the SE results. However, it should be mentioned that the theoretical dielectric function spectra contain more detailed peaks, which are difficult to be distinguished in the experimental dielectric function spectra. This difference can be explained by the fact that in the experimental spectra, some adjacent detailed optical transitions may be combined into one wide-band complex optical transition especially in the high-energy regions. Another reason is that the theoretical calculations depend on the specifically adopted first-principle tools to a certain extent, and different tools are based on different theoretical approximations and consequently produce different details in the calculated results.

Finally, based on the CP analysis on the SE-determined optical functions and first-principle calculated results, we can systematically identify detailed information of the CP optical transitions in the monolayer WS_2_ by using the proposed method. Figure 6a,b, respectively, show the band structure and PDOS of the monolayer WS_2_, where V*_i_* and C*_i_* refer to the *i*-th highest VB and the *i*-th lowest CB, respectively. Figure 6c illustrates the differential band structure *E*_C−V_(*K*), namely the energy difference curves between the three lowest CBs and the three highest VBs, which may be involved in the optical transitions over the concerned energy range. Figure 6d shows the CP analysis results, namely the second derivative of dielectric functions and best-fitting curves, where the energy positions of CPs A–F are precisely determined in the spectra. These four diagrams are displayed in a sequence as arranged in Figure 6, and equal-energy lines (EELs) and equal-momentum lines (EMLs) are used to connect these results, which build a bridge between the SE experiments and the density functional theory to help identify the CP optical transitions of the monolayer WS_2_. Firstly, EELs from the center positions of CPs are drawn, running through the d^2^*ε*/d*E*^2^ curves and the differential band structure diagram. Tangent points between the EELs and the energy difference curves *E*_C−V_ are marked with black solid circles, which can be regarded as the positions in the BZ where the optical transitions occur as shown in Figure 6c. This is because the JDOS at the tangent point is singular, where the transition strength reaches an extreme value. Therefore, the energy and momentum levels of the CP optical transitions can be determined in the BZ *E*-*K* space in this procedure. Then, EMLs from these tangent points are drawn, running through the energy difference curves *E*_C−V_(*K*) and the band structure diagram, and the energy bands including the VBs and CBs involved in the CP optical transitions can be identified. Finally, EELs from the intersection points of the EMLs and the involved energy bands are drawn, running through band structure and PDOS curves, and specific carrier types involved in the CP optical transitions can be determined.

It can be obviously observed from Figure 6 that CP A at 2.019 eV and CP B at 2.410 eV, the widely reported excitonic transitions, are observed due to the direct optical transitions at the high symmetry point K in the BZ from the splitting VBs V_1_ and V_2_ to the lowest CBs C_1_(C_2_) in the HSE06 band structure. The energy difference between CPs A and B is due to SOC effect, and the energy splitting is approximately 0.391 eV, which is consistent with our theoretical calculated value of 0.379 eV as well as with the reported results [61]. The transition position of CP C cannot be identified clearly. We theorize that direct transitions from V_1_(V_2_) to C_1_(C_2_) around the Γ point may induce CP C. Figure 6c shows that CPs D and E can occur in Γ-K or M-Γ zone. The previously published results [4,6,71] have also indicated that the feature peaks of monolayer WS_2_ and MoS_2_ can be associated with the optical transitions between the two highest VBs and the three lowest CBs between the Γ and K points, which are similar to those of CPs D and E in our study. Accordingly, we identify the transition positions of CPs D and E between Γ and K points. Specifically, CP D at 3.033 eV is owing to the direct optical transition from V_1_ to C_1_ at a high symmetry point between Γ and K in the BZ. Similarly, CP E at 3.318 eV is also observed due to the direct optical transition from V_1_ to C_1_ but with a higher energy difference between them at a high symmetry point between Γ and K in the BZ. At last, the high-energy CP F at 4.437 eV is caused by the direct optical transition from V_3_ to C_1_(C_2_) at the high symmetry point K in the BZ. As unambiguously pointed out in the PDOS as shown in Figure 6d, all these CP optical transitions are mainly due to the contributions from the electrons in 3*p* orbitals of S atoms and 5*d* orbitals of W atoms. Therefore, it can be concluded that by using the proposed method, the occurring positions in the BZ and specific energy bands and carrier types involved in CP optical transitions A-F of the monolayer WS_2_ are clearly identified. These results and discussion will provide basic physical knowledge to help us understand the optical and electrical properties of the monolayer WS_2_ and also guide the optimal design of related optoelectronic devices.

It should be noted that the monolayer WS_2_ is only taken as a representative example to demonstrate the effectiveness and advantages of the proposed method. However, the proposed method itself is general, and can be easily extended to identify and interpret the CP optical transitions of other novel 2D semiconductors.

## 4. Conclusions

In summary, a general method is proposed to systematically investigate the optical functions and optical transitions of 2D semiconductors by combining the SE and first-principle calculations. Basic principle and operating procedures of the proposed method are clearly presented in detail. CPs in the experimental optical function spectra determined by the SE are determined by the CP analysis method and connected to first-principle calculated band structure and PDOS via EELs and EMLs, and detailed information about CP optical transitions, including the center energies, occurring positions in BZ, and involved energy bands and carrier types, can be simultaneously identified. In the results and discussion, the CVD single-crystal monolayer WS_2_ is taken as a representative example to demonstrate the effectiveness and advantages of the proposed method. Basic optical functions including the complex refractive index, dielectric functions, and optical conductivity of the monolayer WS_2_ are determined by the SE over an ultra-wide spectral range of 245–1000 nm. Up to six excitonic-type CPs labeled as A-F are quantitatively distinguished in the optical function spectra over the concerned spectral range. By being combinied with the first-principle calculations, these CPs are interpreted as the direct optical transitions involved the three highest VBs (V_1_–V_3_) and three lowest CBs (C_1_–C_3_) at the high symmetry points in the BZ, which are mainly contributed by electrons in S-3*p* and W-5*d* orbitals. These results and discussion provide basic physical knowledge to help us understand and regulate the optical and dielectric properties of the monolayer WS_2_ and also guide the design of related optoelectronic devices.

## Figures and Tables

**Figure 1 nanomaterials-13-00196-f001:**
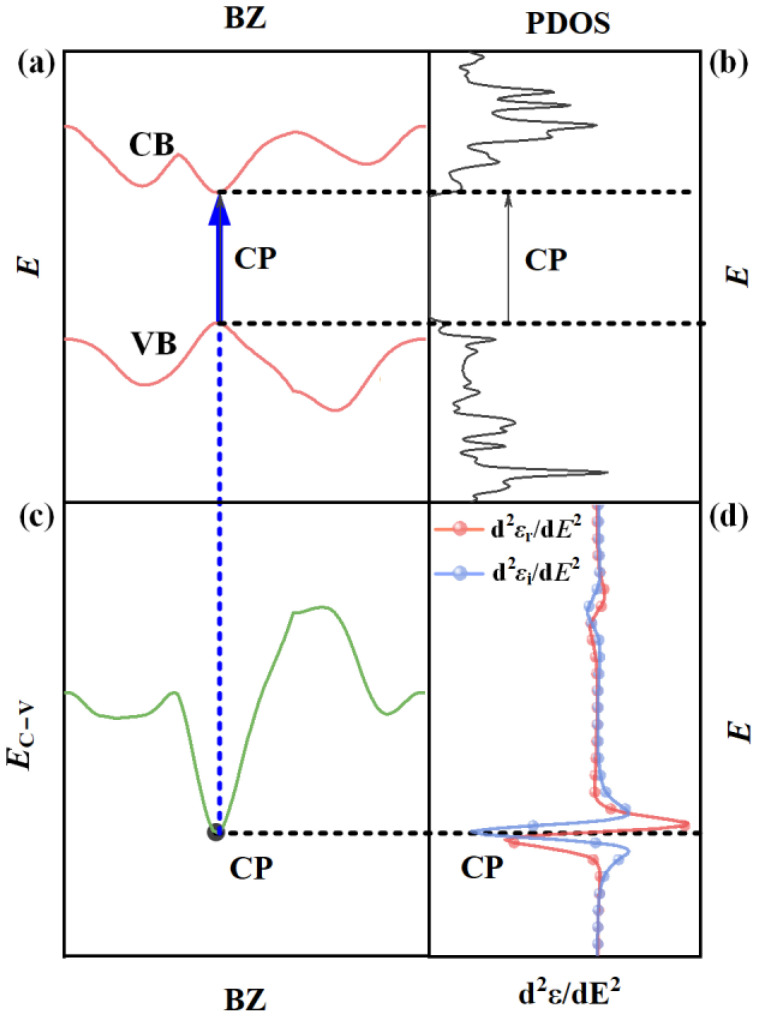
Scheme of the operating principle and procedures of the proposed method to identify the CP optical transitions of 2D materials based on the SE, CP analysis and first-principle calculations: (**a**) first-principle calculated band structure; (**b**) first-principle calculated partial density of states (PDOS); (**c**) differential band structure; (**d**) CP analysis results.

**Figure 2 nanomaterials-13-00196-f002:**
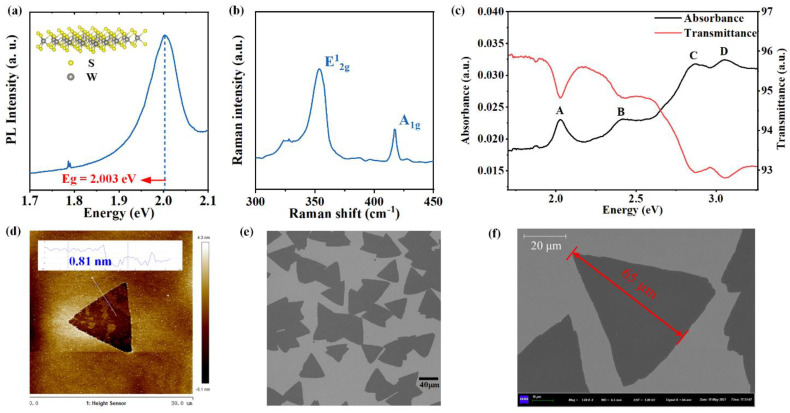
Characterization results of the CVD single-crystal monolayer WS_2_ sample on the sapphire substrate: (**a**) PL spectrum, the inset is a scheme of the lattice structure; (**b**) Raman spectrum; (**c**) absorption and transmission spectra, where uppercase letters A–D represent the feature peaks in the absorption spectrum; (**d**) the AFM image, the inset shows the height curve of the white line across the sample; (**e**,**f**) the SEM images, indicating an example grain size of 65 µm.

**Figure 3 nanomaterials-13-00196-f003:**
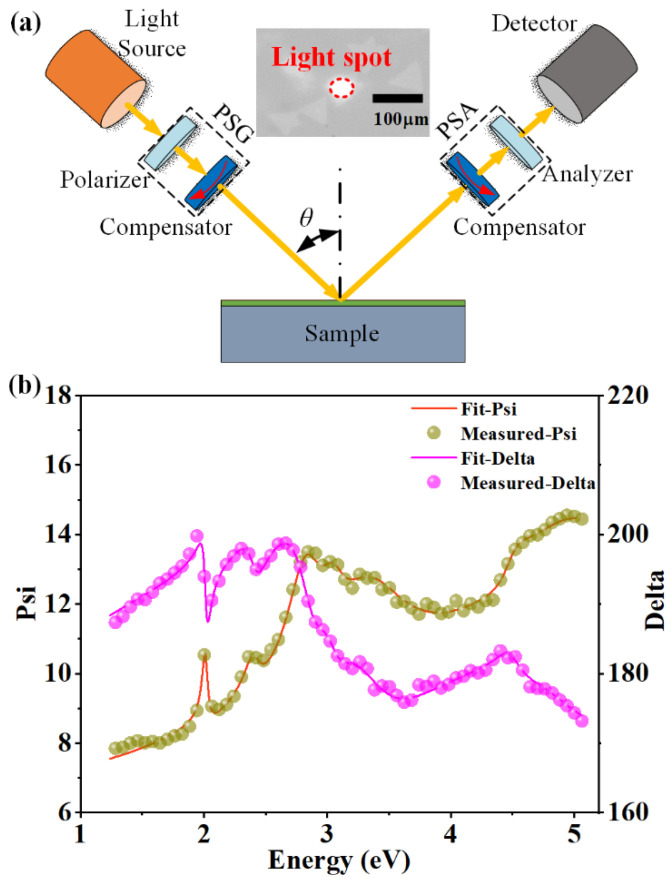
(**a**) Schematic of the spectroscopic ellipsometer, the inset shows the CCD image of light spot irradiated on the sample. (**b**) Measured and best-fitting ellipsometric spectra (Ψ, Δ) of monolayer WS_2_.

**Figure 4 nanomaterials-13-00196-f004:**
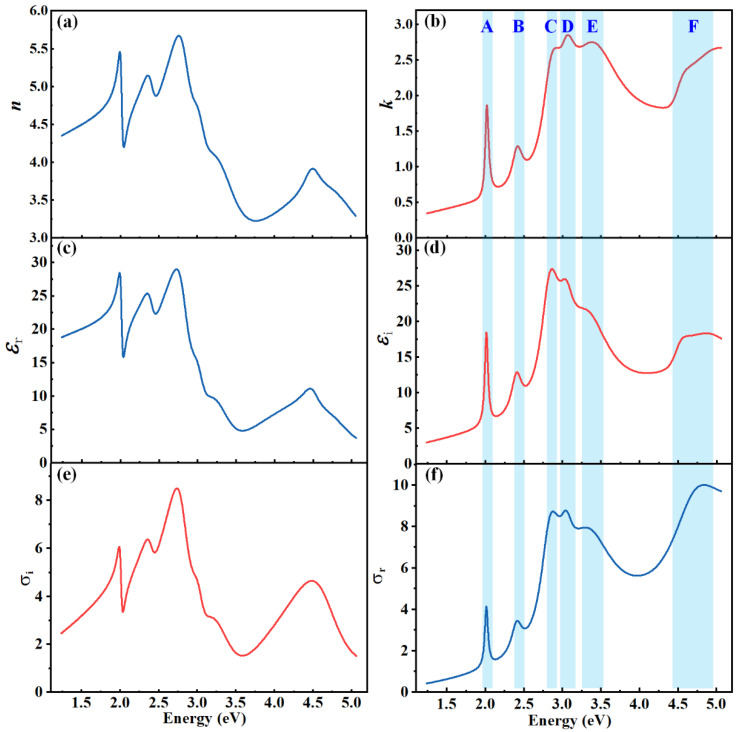
Optical functions of the monolayer WS_2_ determined by the SE: (**a**,**b**) refractive index *n* and extinction coefficient *k*; (**c**,**d**) real part *ε*_r_ and imaginary part *ε*_i_ of the dielectric function; (**e**,**f**) imaginary part *σ*_i_ and real part *σ*_r_ of the optical conductivity. Uppercase letters A–F represent the CPs.

**Figure 5 nanomaterials-13-00196-f005:**
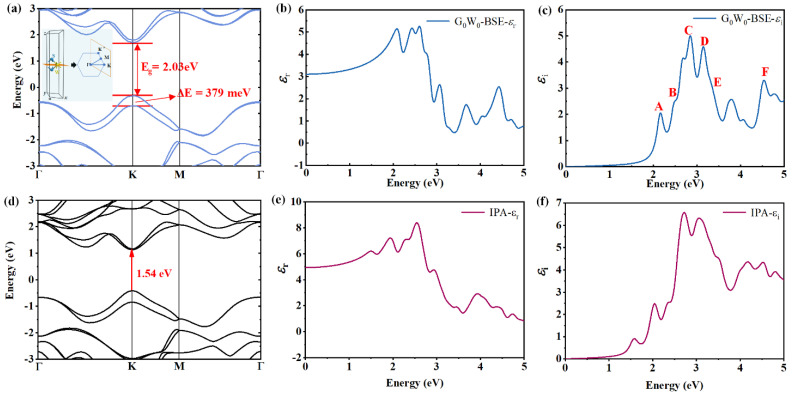
First-principle calculation results of the monolayer WS_2_: (**a**) the HSE06 band structure, the inset shows the Brillouin Zone; (**b**,**c**) real part and imaginary part of the G_0_W_0_ + BSE calculated dielectric function spectra; (**d**) the PBE band structure; (**e**,**f**) real part and imaginary part of the IPA dielectric spectra. Uppercase letters A–F represent the CPs.

**Figure 6 nanomaterials-13-00196-f006:**
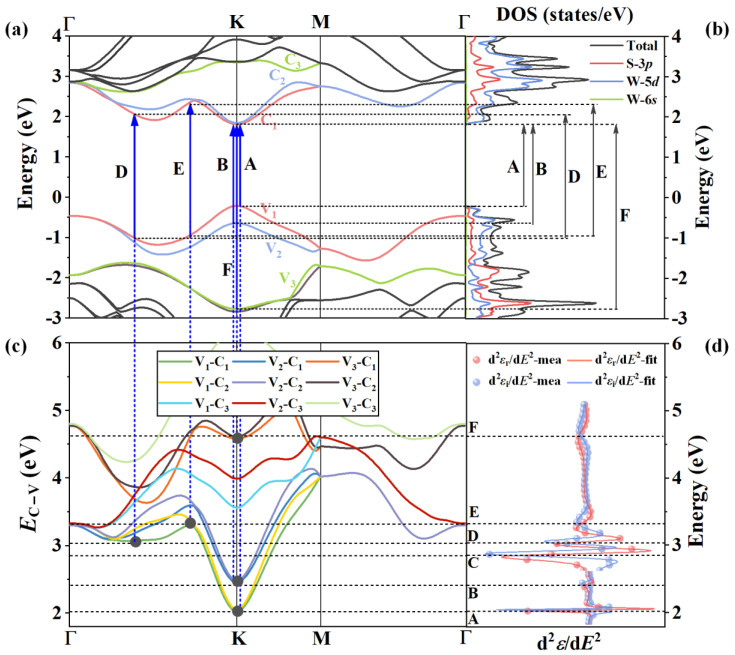
Identification of the CP optical transitions of the monolayer WS_2_: (**a**) first-principle calculated band structure along relevant *k*-branches in the projected Brillouin zone; (**b**) PDOS in the corresponding energy range; (**c**) differential band structure calculated by the energy difference between involved valance and conduction bands in the CP optical transitions; (**d**) CP analysis results, i.e., the second derivative of dielectric functions and best-fitting curves. Uppercase letters A–F represent the CPs. Black dotted lines and blue dotted lines throughout these diagrams are, respectively, the equal-energy lines (EELs) and equal-momentum lines (EMLs). Blue solid lines and black solid lines indicate the CP optical transitions in the band structure and PDOS, respectively. Black solid circles in the differential band structure are the tangent points between the *E*_C−V_(*K*) curves and the EELs, indicating the occurring positions of the CP optical transitions.

**Table 1 nanomaterials-13-00196-t001:** Comparison of CP center energies *E*_0_ obtained from the SE, absorption and transmission (A and T), PL and first-principle calculations.

*E*_0_ (eV)	A	B	C	D	E	F
SE	2.019	2.410	2.844	3.033	3.318	4.438
A and T	2.031	2.428	2.879	3.064	-	-
PL	2.003	-	-	-	-	-
First-principle	2.160	2.495	2.846	3.147	3.352	4.535

**Table 2 nanomaterials-13-00196-t002:** Best-fitting parameters of the CPs analysis for the monolayer WS_2_.

Parameter	A	B	C	D	E	F
*A* (no unit)	0.376	0.423	1.329	1.202	2.971	0.885
*ϕ* (°)	135.087	109.951	91.137	110.086	128.801	114.215
*E*_0_ (eV)	2.019	2.410	2.844	3.033	3.318	4.438
Γ (eV)	0.042	0.098	0.126	0.161	0.330	0.265

## Data Availability

The data presented in this study are available on request from the corresponding author.

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
