# Peer review of "Investigations of Optical Functions and Optical Transitions of 2D Semiconductors by Spectroscopic Ellipsometry and DFT"

_nanomaterials, 2023, doi:10.3390/nano13010196_

Round 1

Reviewer 1 Report

The paper discusses a technique for determining optical functions and optical transitions in 2D materials. The results of detailed studies of WS2 two-dimensional layers by the following methods are presented: PL, absorption/transmission spectroscopy, and ellipsometric spectroscopy. DFT calculations of the WS2 band structure are being carried out. Comparison of the results of calculations and experimental data makes it possible to obtain comprehensive information about the optical functions and optical transitions in the structures under consideration.

However, despite all the advantages of the work, there are a number of remarks to it.

1. I would like the authors to more clearly formulate what is the novelty of their method. As I see it, the authors apply well-known experimental spectral techniques and combine their results with DFT calculations.

2. Figure 1 d - it is not clear what is 2 lines (blue and red)?

3. It is not clear with what radiation wavelength the PL was excited. It is also not clear which radiation was used in the Raman measurements.

4. At what temperatures were the PL and Raman measurements performed? This may be important for the interpretation of the spectra.

5. It is known that DFT calculations are very sensitive to the choice of functional. For example, when trying to ab initio simulate GaAs [AIP Conf. Proc. 1482, 64-68 (2012); doi: 10.1063/1.4757439] the authors of this paper were forced to use different functionals in order to obtain convergence with experiment in terms of the lattice constant and in terms of the band gap. To what extent can this unpleasant circumstance spoil the universality of the proposed approach? Will it not happen that when analyzing new materials, the results will strongly depend on the choice of functional, or will it take an unreasonably long time to select an adequate functional?

6. What caused the broadening of the PL band by 2.003 eV? In the case of purely interband transitions, the PL band should have an asymmetric shape. The high-energy wing of the PL band reflects the temperature distribution of charge carriers in the bands, while the low-energy wing reflects the distribution of fluctuation tails of the density of states. In any case, this is true for bulk materials, such as GaAs [Almuneau, G., Chouchane, F., Calvez, S., Makhloufi, H., Fontaine, C. Three dimensional confinement technology based on buried patterned AlOx layers: Potentials and applications for VCSEL arrays. 15th International Conference on Transparent Optical Networks (ICTON) 2013, pp. 1-3. DOI: 10.1109/ICTON.2013.6602682]. Is it possible that the WS2 PL broadening is caused by inhomogeneous deformations? Is it necessary to take these deformations into account in the authors' method?

7. It is not clear which light source was used in the SE measurements? What is the dependence of the radiation intensity of the source on the quantum energy? This may be important for normalizing measurement results.

8. From the text, the physical meaning of the optical conductivity is not entirely clear. It is desirable to explain it in more detail.

9. Is it clear what caused the difference in the position of the A peak according to the PL, absorption and SE data?

Reviewer 2 Report

The authors describe the procedure of deriving  optical functions of 2D semiconductors from the ellipsometric experimental data, supported by first-principle calculations of the corresponding characteristics of the 2D system.  The core idea has been borrowed from the strategy of interpretation of reflectivity and absorption spectra of bulk semiconductors of insulators exploited several decades ago. However, the authors adopted the idea to the modern ellipsometric techniques of optical measurements of solids and up-to-date tools for first-principle band structure calculations. The procedure gives valuable data concerning the band structure also away from the center of the Brillouin zone. In particular, it helps to assess the energy separation of the occupied and empty electronic bands. Application of it is presented for WS2, one of the intensively investigated transition metal dichacogenides.  The procedure is well described and the paper can be interesting for a large community studying 2D semiconductors.

I would suggest to make the title more precise. In its present form, it is to general and promises to much. The manuscript refers to a particular experimental technique (spectroscopic ellipsomery) and particular methods of DFT-based calculations.    

Some references would be useful in the part describing “forward modeling procedure” (starting from the line 136).

The first author of Ref. 51 is Peter Y. Yu, so his last name is Yu.

Some typos should be fixed.

Round 2

Reviewer 1 Report

Authors have clarified all comments. The paper can be published after easy minor revision. Some simple unclear moments are listed bellow.

Comment 2

As authors write at the line 141, real and  imaginary  part of the dielectric function are ɛr and ɛi, respectively. It will be better to mark them on fig. 1d and in the text uniformly.

Comment 5.

What is "DC"? It is not clarified in the text of the paper (and in the text of the response, unfortunately...).  

Comment 6.

I would suggest to say here not about assymetric shape of PL band, but about inhomogeneous broadering of PL band. Yes, it can be caused by substrate factor (by deformation in particular).

Comment 7.

It would be good to clarified SE measurements detail in the text of the paper.

Comment 8.

The simbol "-" is not correct at reference at line 382 (65-66). The "," will be better.

Comment 9.

It will be better to extend this explanation to all peaks. The differencies in A-T peaks position and SE peaks position for high-energy peaks is similar to A peak.

Moreover, it will be good to check abstract volume after revison to satisfy the Journal rules.
